# Fabrication and Characterization of Zein Composite Particles Coated by Caseinate-Pectin Electrostatic Complexes with Improved Structural Stability in Acidic Aqueous Environments

**DOI:** 10.3390/molecules24142535

**Published:** 2019-07-11

**Authors:** Yaqiong Zhang, Bo Wang, Yan Wu, Boyan Gao, Liangli (Lucy) Yu

**Affiliations:** 1Department of Food Science & Engineering, School of Agriculture and Biology, Shanghai Jiao Tong University, Shanghai 200240, China; 2Department of Nutrition and Food Science, University of Maryland, College Park, MD 20742, USA

**Keywords:** zein, caseinate, pectin, composite particles, curcumin, acidic stability

## Abstract

Zein composite particles coated with caseinate-pectin electrostatic complexes (zein-caseinate-pectin particles) were fabricated using an electrostatic deposition and liquid-liquid dispersion method without heating treatment. Compared to zein particles coated only with caseinate, the acidic stability of zein-caseinate-pectin particles was greatly improved, and the particle aggregation was suppressed at pH 3–6, especially at pH values near the isoelectric point of caseinate (pH 4–5). Besides, desirable long-term storage stability and re-dispersibility were observed. Under different zein to curcumin (Cur) feeding ratios (10:1, 20:1, 30:1 and 40:1, *w*/*w*), the Cur-loaded zein-caseinate-pectin particles had a spherical shape with an average diameter ranging from 358.37 to 369.20 nm, a narrow size distribution (polydispersity index < 0.2) and a negative surface charge ranging from −18.87 to −19.53 mV. The relatively high encapsulation efficiencies of Cur (81.27% to 94.00%) and desirable re-dispersibility were also achieved. Fluorescence spectroscopy indicated that the encapsulated Cur interacted with carrier materials mainly through hydrophobic interactions. The *in-vitro* release profile showed a sustained release of Cur from zein-caseinate-pectin particles in acidic aqueous environment (pH 4) up to 24 h, without any burst effect. In addition, the encapsulation retained more ABTS^•+^ radical scavenging capacity of Cur during 4 weeks of storage. These results suggest that zein-caseinate-pectin particles may be used as a potential delivery system for lipophilic nutrients in acidic beverages.

## 1. Introduction

Zein is the major storage protein in corn with high hydrophobicity, desirable biodegradability and biocompatibility [1,2]. Recently, zein particles have been investigated in food and pharmaceutical industries for delivering lipophilic nutrients and pharmaceuticals, such as fish oil [3], curcumin [4], α-tocopherol [5], and 5-fluorouracil [6]. However, due to its high surface hydrophobicity, bare zein particles have relatively poor stability in aqueous systems and undesirable re-dispersibility after drying [7]. This problem can be partly overcome by coating zein particles with protein-based emulsifiers, such as β-lactoglobulin [8] and sodium caseinate [9]. Our previous study showed that zein-caseinate particles had desirable physical stability and re-dispersibility in a neutral aqueous environment, which may be due to the enhanced electrostatic and steric stabilization by caseinate molecules adsorbed on the surface of zein particles [10,11]. Nevertheless, zein-caseinate particles are still highly sensitive to aggregation at pH values near or below pH 4.6, the isoelectric point of the adsorbed caseinate molecules, because under these conditions the electrostatic repulsion between the particles is weakened and insufficient to overcome the attractive interactions [12].

Recently, the protein and polysaccharide complexes have been used as the coating layer for zein particles and the formed particles have shown an improved physical stability in acidic aqueous environment by having a thicker coating layer and stronger steric repulsion [13]. Some previous studies have shown that the coating of caseinate-pectin complex particles could enhance the stability of zein particles under simulated gastrointestinal conditions and meanwhile improve their encapsulation capacity [14,15]. Nevertheless, the fabrication for zein-caseinate-pectin complex particles was achieved via a pH- and heating-induced electrostatic adsorption process. This approach for complex particle formation relies on the thermal denaturation and aggregation of protein molecules as well as electrostatic interactions, which may be affected by many factors, including the heating temperature, pH, charge density, ionic strength, as well as biopolymer concentrations and ratios [16].

Thus, the primary purpose of this study was to fabricate zein composite particles stabilized by caseinate-pectin complexes using an electrostatic deposition and liquid-liquid dispersion method without heating treatment, and the acidic stability of zein-caseinate-pectin composite particles was also investigated. Moreover, curcumin was selected as a lipophilic probe compound because of its hydrophobicity, environmental instability and potential health benefits. Curcumin-loaded zein-caseinate-pectin composite particles were characterized for their important physico-chemical features, including size, size distribution, surface charge, morphology, encapsulation efficiency intermolecular interaction, and *in-vitro* release behavior. Finally, the *in-vitro* antioxidant activity of the encapsulated curcumin incorporated into an acidic model beverage was evaluated and compared with that of the free curcumin.

## 2. Results and Discussion

### 2.1. Preparation and Characterization of Zein-Caseinate-Pectin Particles

#### 2.1.1. Influence of pH

Initially, the influence of pH on the electrical characteristics of caseinate and pectin aqueous solutions was examined from ζ-potential measurements (Appendix A). The ζ-potential of caseinate molecules changed from highly positive (+25.7 mV) at pH 3 to highly negative (−36.8 mV) at pH 7, with the point of zero charge around pH 4.5, which was similar to that observed in a previous study [17]. The ζ-potential of pectin molecules remained highly negative from pH 3 to 7, with an increase in the magnitude of the negative charge from −24.7 to −50.5 mV, which can be attributed to partial deprotonization of the charged carboxyl groups [18]. Since caseinate and pectin aqueous solutions have opposite net charges in the pH range of 3 to 4, pH 3.5 was chosen to facilitate the formation of caseinate-pectin electrostatic complexes.

#### 2.1.2. Influence of Mass Ratio of Pectin to Caseinate

At pH 3.5, a group of caseinate-pectin electrostatic complexes containing different pectin to caseinate mass ratios were first prepared and their ζ-potential values were measured, respectively (Figure 1). At the pectin to caseinate mass ratio of 0.2, many large and small aggregates were formed instantaneously after mixing these two solutions. This result suggested that the complexes formed under this condition were highly unstable, possibly due to their extremely low surface potential of +0.2 mV (Figure 1). With increasing the pectin to caseinate mass ratio from 0.2 to 1, relatively stable caseinate-pectin complexes were formed, and no obvious aggregates and sediments were observed. The formed complexes were negatively charged and had a continuous increase in the magnitude of the negative charge to −31.1 mV at the pectin to caseinate mass ratio of 1. Further increasing pectin to caseinate mass ratio from 1 to 2, the ζ-potential value was almost not changed and similar to that of pectin alone at pH 3.5 (−28.2 mV, Appendix A), indicating the saturation of pectin molecules [18].

Then, zein-caseinate-pectin particles containing the constant concentration of zein were produced with caseinate-pectin complexes with different pectin to caseinate mass ratios and their ζ-potential values were also measured (Figure 1). In the absence of caseinate-pectin complexes, the ζ-potential value of zein particles was +25.6 mV, which was consistent with a previously published finding [7]. Caseinate-pectin complexes had a negative charge (at the pectin to caseinate mass ratio higher than 0.2) and were likely to adsorb to the surfaces of oppositely charged zein particles. As shown in Figure 1, the ζ-potential reversal of zein particles after mixing with caseinate-pectin complexes from positive to negative, indicated the successful adsorption of caseinate-pectin complexes onto zein particle surfaces. Compared to that of caseinate-pectin complexes at the same pectin to caseinate mass ratio, a less absolute negative ζ-potential value was observed for zein-caseinate-pectin particles, which might be attributed to possible charge neutralization by the cationic groups in zein molecules [13]. Moreover, it was observed that a limiting ζ-potential value (−20.5 mV) of zein-caseinate-pectin particles was also reached at pectin to caseinate mass ratio of 1 and further increasing its mass ratio to 2 did not significantly change the ζ-potential value (−20.4 mV). Therefore, pectin to caseinate mass ratio of 1 was used in the following experiments.

#### 2.1.3. The Stability of Zein-Caseinate-Pectin Particles at Different Acidic pH Values

The stabilities of zein-caseinate-pectin particles at selected acidic pH values (pH 3–6) were first studied through visual inspection for the occurrence of aggregation or sediments. As shown in Figure 2A, zein-caseinate-pectin particles had desirable stability across the entire tested pH range, and no obvious particle aggregates and sediments were observed, even at pH values (pH 4–5) around the isoelectric point of caseinate. In contrast, there was an extensive particle aggregation phenomenon for zein-caseinate particles without adding pectin at pH 4, 4.5 and 5 (Figure 2B). These results suggested that the coating of caseinate-pectin complexes greatly improved the acidic stability of zein particles, especially at the pH values near the isoelectric point of the caseinate.

In addition, the particle size, particle distribution and ζ-potential values of zein-caseinate-pectin particles at different acidic pH values were measured (Table 1). With the increase of pH value from 3 to 6, a statistically significant increase of the particle size from 291.6 to 405.7 nm was observed (*p* < 0.05). Meanwhile, a significant increase in the magnitude of the negative ζ-potential value from −9.8 to −37.6 mV was observed (*p* < 0.05). According to the previous study, the pKa of pectin ranges between 3 and 4, depending on the type and source of pectin [19]. Therefore, an increase in pH value from 3 to 6 enhances the hydration degree of pectin molecules, due to the easier penetration of water molecules into carboxylate anions (pectin salt) than free carboxyl groups (pectinic acid), and results in an increased hydrodynamic diameter and negative ζ-potential value of zein-caseinate-pectin particles. All particle samples had a polydispersity index (PDI) around 0.2, indicating a narrow size distribution and good aggregation stability of the particles.

#### 2.1.4. Re-Dispersibility

The re-dispersibility of zein-caseinate-pectin particles was determined and the particle size, particle distribution and ζ-potential values were compared before and after lyophilization (Table 1). Zein-caseinate-pectin particles were re-dispersed in water quickly after gentle shaking, forming uniform and milky dispersion systems (data not shown). The desirable re-dispersibility of particles at acidic pH values re-confirmed the effective coverage of caseinate-pectin complexes on the surfaces of zein particles, which provided both electrostatic and steric protections, known as electrosteric stabilization [20]. As shown in Table 1, the re-dispersed zein-caseinate-pectin particles had larger particle sizes than those of the freshly prepared samples. The particle sizes were in the range of 291.6 and 405.7 nm before lyophilization and increased to the range between 385.0 and 573.1 nm after re-dispersion. However, a narrow particle size distribution (PDI around 0.2) of samples after re-dispersion was obtained. Moreover, the observed higher magnitudes of the negative ζ-potential values of zein-caseinate-pectin particles after re-dispersion indicated their desirable aggregation stability [21], which was consistent with the particle size distribution result.

#### 2.1.5. Storage Stability

The storage stability of zein-caseinate-pectin particles was examined for their changes in particle size, size distribution and ζ-potential value after storing at 4 °C for 8 weeks (Figure 3). Compared to those of the freshly prepared particles at a certain pH value, no significant change in the particle size was observed throughout 8 weeks of the study, except for the zein-caseinate-pectin particles at pH 3 (Figure 3A). A significant decrease (*p* < 0.05) of particle size from 291.6 to 261.1 nm after 8 weeks storage might be due to its relatively low magnitude of ζ-potential value (around −10 mV) (Figure 3C). This finding was supported by Koutsoulas’s observation that for combined electrostatic and steric stabilized colloidal dispersions, the minimum ζ-potential for particle stabilization should be ±20 mV [22]. Besides, polydispersity indexes (PDI) all remained within the range of acceptably monodisperse (<0.2) (Figure 3B). As shown in Figure 3C, the ζ-potential values for zein-caseinate-pectin particles did not change appreciably during storage. Except for the zein-caseinate-pectin particles at pH 3, the absolute ζ-potential values for particles were all close to or above 20 mV, which suggested good stability under the pH conditions.

### 2.2. Preparation and Characterization of Cur-Loaded Zein-Caseinate-Pectin Particles

Cur-loaded zein-caseinate-pectin particle dispersions were prepared using different zein to Cur mass ratios, and the visual appearances of different particle dispersions are shown in Appendix A. All the particle dispersions were optically translucent yellow and the most intense yellow color occurred for the dispersion with zein to Cur mass ratio of 10:1. As shown in Table 2, the particle size, polydispersity index, and ζ-potential of the Cur-loaded zein-caseinate-pectin particles were not significantly affected by the different zein to Cur mass ratios (*p* > 0.05). These results suggested that Cur was encapsulated inside the particles and did not have an obvious impact on the formation or surface properties of particles. The encapsulation efficiencies of Cur were relatively high (81.3–94.0%), with the encapsulation efficiency increasing significantly with increasing zein to Cur mass ratio from 10:1 to 40:1 (*p* < 0.05). Compared to the very low solubility of Cur in water (0.6 μg/mL) [23], the encapsulated Cur in zein-caseinate-pectin particles had a greatly increased solubility of 23.5–81.3 μg/mL, indicating a 39–135 fold increase in water solubility of Cur.

### 2.3. Re-Dispersibility

When the freeze-dried yellow zein-caseinate-pectin particle powder was re-dispersed in water, the resulting particle dispersion had a similar appearance as the freshly prepared sample and no obvious visible aggregates or sediments were detected (data not shown). Furthermore, consistent with those of the blank zein-caseinate-pectin particles, the re-dispersed Cur-loaded particles also showed larger particle sizes and higher magnitude of the negative charges than those of the freshly prepared counterpart samples. The particle size distribution of samples after re-dispersion did not change and all the polydispersity indexes (PDI) were still around 0.2, indicating a narrow particle size distribution of re-dispersed particles (Table 2).

### 2.4. TEM Study

Transmission electron microscopy (TEM) analysis showed that the zein-caseinate-pectin particles were spherical, which was well dispersed or partly adhered (Figure 4). Most of the particles in the TEM images had diameters in the range 100–220 nm, but there were also some smaller particles. The particle size observed by TEM was therefore somewhat smaller than the hydrodynamic diameter as determined by dynamic light scattering (DLS), which showed an average particle diameter around 360 nm (Table 2). This may be explained by the fact that most of the water in the particles was evaporated during the drying process required for TEM visualization, which resulted in a particle shrinkage. A similar observation was also reported in a previous study on the zein particles coated by pectin [24].

### 2.5. Fluorescence Study

Intrinsic fluorescence of proteins had been extensively utilized to study the binding properties of proteins with other small molecules [25,26]. According to the previous study, free Cur showed a little fluorescence signal at the range 300–450 nm, which did not interfere with protein fluorescence [27]. Under an excitation wavelength of 280 nm, blank and Cur-loaded zein-caseinate-pectin particle dispersions all showed the maximum emission peak of the fluorescence spectra about 336 nm (Figure 5A), which may be contributed to the hydrophobic tryptophan residue in proteins [9]. With increasing the Cur loading, the fluorescence intensity of particle dispersions decreased progressively without changing the shape of emission maximum. All these results indicated that Cur could quench the intrinsic fluorescence of proteins, referring to the decreased fluorescence intensity because of molecular interaction.

The quenching of intrinsic fluorescence was further investigated using the Stern–Volmer equation, and the ratio of fluorescence emission intensities without or with Cur (F_0_/F) vs. Cur concentration was linearly related (R^2^ > 0.98) (Figure 5B). The Ksv and kq were calculated to be 1 × 10^4^ M^−1^ and 0.3 × 10^12^ M^−1^s^−1^, respectively. The kq was much greater than the maximum dynamic quenching constant (2 × 10^10^ M^−1^s^−1^), indicating that Cur-induced fluorescence quenching was due to the static quenching, corresponding to the formation of complexes between particle proteins (mainly hydrophobic amino acid residue) and Cur [28].

### 2.6. In-vitro Release Behavior

The release profiles of free Cur and encapsulated Cur from zein-caseinate-pectin particles were shown in Figure 6. A fast diffusion profile was observed for free Cur with around 40% being diffused within the first 2 h and almost all the remaining Cur diffused at the end of 4 h. In contrast, when Cur was encapsulated into zein-caseinate-pectin particles, it showed a much slower release behavior in acidic release medium. The cumulative release of encapsulated Cur was found to be 2.2% after 2 h, 17.2% after 4 h, 37.4% after 6 h, 54.3% after 8 h, 66.9% after 10 h, 75.0% after 12 h and ultimately 96.8% at the end of 24 h, respectively. The gradual sustained release profile without obvious burst release of Cur from zein-caseinate-pectin particles was observed, indicating an efficient encapsulation of Cur inside the particles and little Cur adsorbed on or near the surfaces of particles. This result was consistent with the relatively high encapsulation efficiency of Cur in Table 2. Besides, a previous literature also showed that a secondary pectin coating on protein nanoparticles could obviously decrease the burst release of encapsulated compound in the acidic condition [29].

### 2.7. Application in Model Beverage and Evaluation of Antioxidant Activity

The antioxidant activity of Cur is well known [30,31]. In this study, antioxidant activity of the encapsulated Cur was investigated and compared with the free Cur in an acidic model beverage system by applying ABTS assay (Figure 7). For the model beverage system incorporated with freshly prepared Cur-loaded zein-caseinate-pectin particles, it showed a significantly higher antioxidant activity than that of the beverage system incorporated with free Cur (*p* < 0.05). Besides, the beverage system incorporated with blank zein-caseinate-pectin particles also showed the antioxidant activity, which might be attributed to the ABTS^•+^ radical scavenging ability of proteins and polysaccharide [32,33]. Levine et al. also showed that some amino acid residues such as methionine and cysteine in caseins or casein-derived peptides can be oxidized by free radicals [34]. Moreover, the ABTS^•+^ radical scavenging ability of the beverage system incorporated with free Cur was almost lost after only one week’s storage at 4 °C, since its radical scavenging capacity obviously decreased from 5.8% to 1.0%. By contrast, the encapsulated Cur incorporated system maintained a relatively stable ABTS^•+^ radical scavenging ability ranging from 17.0% to 21.0% during the four weeks’ storage. This result was similar to the visual observation of these two beverage systems. As shown in Figure 8, the beverage system incorporated with the encapsulated Cur remained translucent yellow even after 4 weeks storage, while the system incorporated with free Cur almost lost its yellow color and became colorless. Previous studies have shown that free Cur was highly sensitive to various environmental stresses, e.g., light, heat or oxygen and its chemical stability can be greatly improved after encapsulating into zein particles [4,10]. Moreover, based on the release profile of encapsulated Cur, a sustained release profile was observed in the release medium of acidic citrate buffer (pH 4) (Figure 6), which was the same as the main component of model beverage system. Therefore, the observed higher antioxidant activity of the encapsulated Cur in this study might be due to the facts that zein-caseinate-pectin particles not only partly protected the encapsulated Cur from degradation but also facilitated its dispersion and resulted in a sustained release from the particles when incubated in an aqueous environment [35].

## 3. Materials and Methods

### 3.1. Materials

Zein with a minimum protein content of 97% was obtained by Showa Sangyo (Tokyo, Japan). Low methoxyl pectin from citrus peel (degree of esterification 15%) was kindly provided by Professor Yapeng Fang’s research group (Shanghai Jiao tong University, Shanghai, China). Sodium caseinate (caseinate), Curcumin (Cur, ≥99.5% purity) were purchased from Sigma-Aldrich Chemical Co., Ltd. (St. Louis, MO, USA). All other reagents were of analytical grade and used without further purification. Water purified with a Milli-Q system with resistivity of 18.2 mΩ was used for all experiments.

### 3.2. Preparation of Zein-Caseinate-Pectin Particles

Caseinate and pectin aqueous solutions were separately prepared and then adjusted to pH 3.5 using HCl. The caseinate aqueous solution (1 mg/mL) was added slowly to the same volume of the pectin aqueous solution with concentrations ranging from 0.2 to 2 mg/mL under continuous stirring using a magnetic stirrer (IKA^®^C-MAG HS 7, IKA WORKS Inc., Wilmington, NC, USA) at ambient temperature. The pectin to caseinate mass ratio used throughout this study was from 0.2:1 to 2:1 (*w*/*w*).

Zein was dissolved in 80 mL/100 mL aqueous ethanol solutions to obtain a stock solution with the final concentration of 10 mg/mL. For preparation of zein-caseinate-pectin particles, a liquid-liquid dispersion method was used following a laboratory procedure [11] with some slight modifications. Briefly, 1 mL of zein stock solution was added slowly into 10 mL of the diluted caseinate-pectin complex dispersions (pH 3.5) with continuous stirring for 30 min at ambient temperature. Then, N_2_ was flowed to remove the ethanol from the system using a nitrogen evaporator (N-EVAPTM 112, Organomation Associates Inc., Berlin, MA, USA). The final dispersions were centrifuged at 4500 rpm for 5 min to remove any large particles and then adjusted to a series of acidic pH values by adding HCl or NaOH.

### 3.3. Characterization of Zein-Caseinate-Pectin Particles

#### 3.3.1. Particle Size, Size Distribution and Ζeta-Potential

The particle size, size distribution and ζ-potential of particle samples were determined for dynamic light scattering (DLS) capacity using a Zetasizer Nano ZS90 (Malvern Instruments Ltd., Worcestershire, UK). The DLS measurements were done with a He-Ne laser wavelength of 633 nm and an angle detection of 90°. Reflective index and viscosity of water were 1.590 and 0.8904 cP, respectively, which were used for calculating effective diameter from autocorrelation. All measurements were carried out at 25 °C and the results were the average of three readings.

#### 3.3.2. Re-Dispersibility

The freshly prepared particle dispersions were frozen at −80 °C overnight and freeze-dried for 48 h to obtain dry powders using a Labconco Freezone 6 plus (Labconco Corp., Kansas City, MO, USA). The freeze-dried samples were re-dispersed in Milli-Q water to the original concentration, and the re-dispersibility was first studied through visual inspection for the occurrence of aggregation. Samples without visible aggregation were subjected to further size, size distribution and ζ-potential characterizations.

#### 3.3.3. Storage Stability

The freshly prepared particle dispersions were kept in a screw-capped clear glass vial and kept in a refrigerator at 4 °C over a period of 8 weeks. The samples were characterized for changes in particle size, size distribution or ζ-potential at the designated sampling time points: 0, 2, 4, 6 and 8 weeks of storage.

### 3.4. Preparation of Cur-Loaded Zein-Caseinate-Pectin Particles

Zein and Cur were co-dissolved in 80 mL/100 mL aqueous ethanol solutions with different zein to Cur mass ratios (10:1, 20:1, 30:1 and 40:1, *w*/*w*) to obtain four stock solutions. The final concentration of zein in stock solutions was fixed at 10 mg/mL, while the concentrations of Cur ranged from 0.25 to 1 mg/mL. Cur-loaded zein-caseinate-pectin particles were also prepared following the procedure described in Section 3.2.

### 3.5. Characterization of Cur-Loaded Zein-Caseinate-Pectin Particles

#### 3.5.1. Encapsulation Efficiency (EE)

The freshly prepared Cur-loaded zein-caseinate-pectin particles were centrifuged at 4500 rpm for 30 min to remove any large particles and non-encapsulated Cur. The obtained insoluble sediment was washed with 1 mL of ethanol and vortexed for 1 min. The resulting solution was then centrifuged at 4500 rpm for 10 min to remove any insoluble matter. The washing procedure was repeated three times and the resulting supernatant was collected and combined. The amount of non-encapsulated Cur in the supernatants was measured according to a laboratory protocol and subjected to ultra-performance liquid chromatography (UPLC) (Waters, Milford, MA, USA) analysis [10]. The SHIMADZU UPLC-30A system was equipped with a SPD-M20A diode array detector and a DGU-20A_5_ prominence degasser. A Waters ACQUITY UPLC BEH C_18_ column (100 mm × 2.1 mm i.d., 2 μm) (Waters, Milford, MA, USA) was used. Acetonitrile and trifluoroacetic acid (0.1%, *w*/*v*) at a volume ratio of 50:50 were used as the mobile phase in an isocratic mode at a flow rate of 0.3 mL/min. The sample injection volume was 5 μL, and the detection wavelength was 425 nm. The calibration curve previously established (Y = 89511 × C +22784, R^2^ = 0.999) was used to determine the Cur concentration based on the sample peak area. The encapsulation efficiency (EE) of Cur was estimated as the percentage of Cur encapsulated in zein-caseinate-pectin particles by the following equation:

EE (%) = [1 − amount of non-encapsulated Cur/total amount of added Cur] × 100
(1)


#### 3.5.2. Re-Dispersibility

The re-dispersibility of Cur-loaded zein-caseinate-pectin particles was also examined following the procedure described in Section 3.3.2.

#### 3.5.3. Transmission Electron Microscopy (TEM)

The morphology of the Cur-loaded zein-caseinate-pectin particles was analyzed by transmission electron microscopy (TEM) photographs using a Tecnai^TM^ G^2^ Spirit Biotwin (FEI Company, Hillsboro, OR, USA), operating at 120 kV. The particle dispersion was diluted using Milli-Q water and one drop of the diluted dispersion was placed on a 200-mesh carbon-coated copper grid. The photographs were taken at the selected magnifications and the representative images were reported.

#### 3.5.4. Fluorescence Spectroscopy

Intrinsic fluorescence of particle samples with or without Cur was recorded by a TECAN Infinite M200 PRO multilabel plate reader (Tecan Group Ltd., Mannedorf, Switzerland). The excitation wavelength was set at 280 nm and emission signal was collected from 300 to 450 nm. The excitation and emission slit widths were fixed at 5 nm.

To explore the fluorescence quenching mechanism, the quenching of intrinsic fluorescence was further investigated using the Stern–Volmer equation:

F_0_/F = 1 + Ksv [Q] = 1 + Kqτ_0_[Q]
(2)
where F_0_ and F are fluorescence intensities without or with a quencher (Cur), respectively; [Q] is the concentration of Cur (M); Kq is the biomolecular quenching rate constant, M^−1^s^−1^; τ_0_ is the lifetime of fluorophore fluorescence in the absence of Cur and equals 3 × 10^−8^ s [9]; and Ksv is the Stern–Volmer quenching constant, which can be determined by the linear slope of the F_0_/F plot against [Q], M^−1^.

#### 3.5.5. *In-vitro* Release Behavior

*In-vitro* release behavior of Cur from zein-caseinate-pectin particles was examined using the dialysis method as previously reported with slight modifications [36]. To simulate the acidic aqueous environment, the Cur release was determined in 0.1 M citrate buffer at pH 4. As the aqueous solubility of Cur was very low, 1% (*w*/*v*) Tween 80 was added into the release medium to achieve the sink condition. Briefly, 20 mL of Cur-loaded zein-caseinate-pectin aqueous dispersion (Cur concentration of 10 μg/mL) was placed into a dialysis bag (Molecular Weight Cut Off of 15kDa), which was then immersed in a conical flask containing fresh release media. The conical flask was incubated in a bath at 37 °C with a shaking speed of 60 rpm. Aliquot (200 μL) of the release medium was collected at the selected time intervals and an equal volume of fresh medium was added each time. The concentration of Cur released into the medium was quantitatively analyzed using the UPLC method as described in Section 3.5.1. For comparison, the free Cur solution (dissolved in 80 mL/100 mL aqueous ethanol solution) with the same Cur concentration was placed into the dialysis bag and its release behavior was also determined by the same procedure. 

### 3.6. Application in Model Beverage and Evaluation of Antioxidant Activity

The model beverage consisted of 0.1 M citrate buffer at pH 4, 13% sucrose and 0.025% sodium benzoate [37]. The freshly prepared Cur solution (dissolved in 80 mL/100 mL aqueous ethanol solutions) and Cur-loaded zein-caseinate-pectin particle dispersion was separately mixed with the model beverage to have a same final Cur concentration of 10 μg/mL. Besides, the blank particle dispersion was mixed with the model beverage and its concentration was the same as that in the Cur-loaded particle dispersion. The obtained beverage systems were kept in a refrigerator at 4 °C and samples were removed at the designated time points. The antioxidant activity of the sample was evaluated by the radical scavenging capacity against ABTS^•+^ according to a laboratory protocol [38]. The scavenging capacity was calculated using the following equation:

ABTS scavenging capacity (%) = [1 − A_s_/A_c_] × 100
(3)
where A_c_ is the absorbance of the control and A_s_ is the absorbance of the different sample.

### 3.7. Statistical Analysis

Data are reported as the mean ± SD for triplicate determinations. One-way ANOVA and Tukey’s test were employed to identify differences in means. Statistics were analyzed using the SPSS for Windows (version rel. 10.0.5, 1999, SPSS Inc., Chicago, IL, USA). Statistical significance was declared at *p* < 0.05.

## 4. Conclusions

Zein composite particles coated by caseinate-pectin electrostatic complexes were successfully prepared and showed improved stability in acidic aqueous environments. Curcumin was encapsulated into zein-caseinate-pectin composite particles mainly through hydrophobic interactions. This study demonstrated that zein-caseinate-pectin composite particles hold promising potential as a delivery system for lipophilic nutrients, especially in acidic functional beverages and drinks.

## Figures and Tables

**Figure 1 molecules-24-02535-f001:**
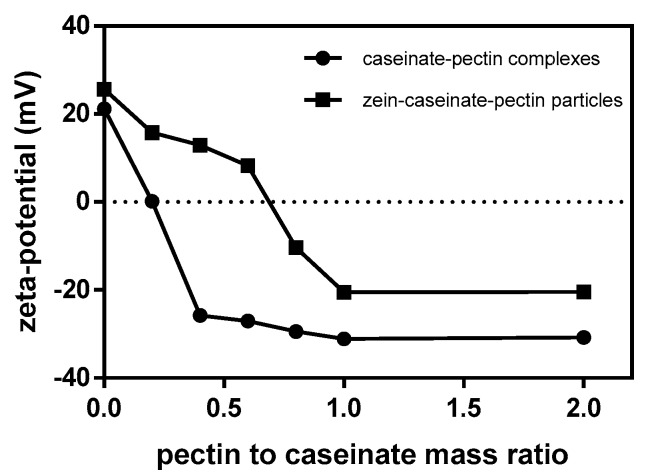
ζ-potential as a function of pectin to caseinate mass ratio of caseinate-pectin complexes (●) and zein-caseinate-pectin particles (■).

**Figure 2 molecules-24-02535-f002:**
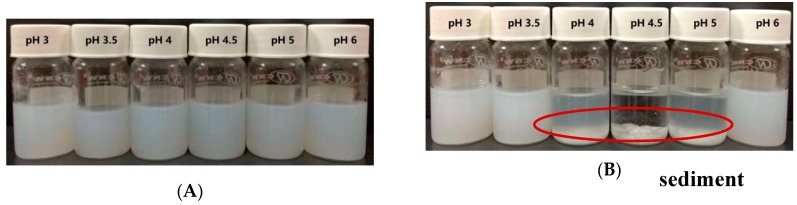
Visual appearance of zein-caseinate-pectin particles (**A**) and zein-caseinate particles (**B**) at different acidic pH conditions.

**Figure 3 molecules-24-02535-f003:**
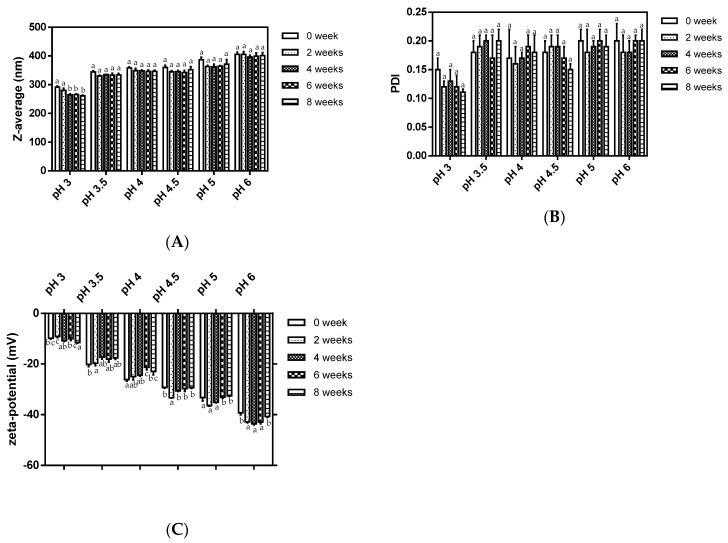
Z-average (**A**), polydispersity index (PDI) (**B**) and ζ-potential (**C**) changes of zein-caseinate-pectin particles during storage at 4 °C.

**Figure 4 molecules-24-02535-f004:**
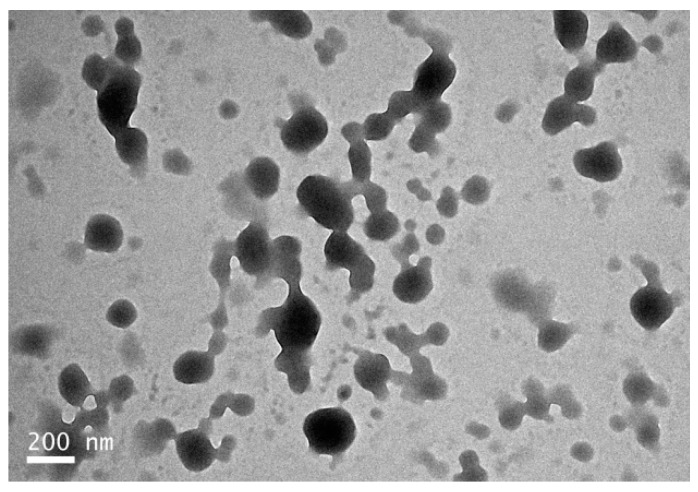
Transmission electron microscopy (TEM) image of the freshly-prepared Cur-loaded zein-caseinate-pectin particles.

**Figure 5 molecules-24-02535-f005:**
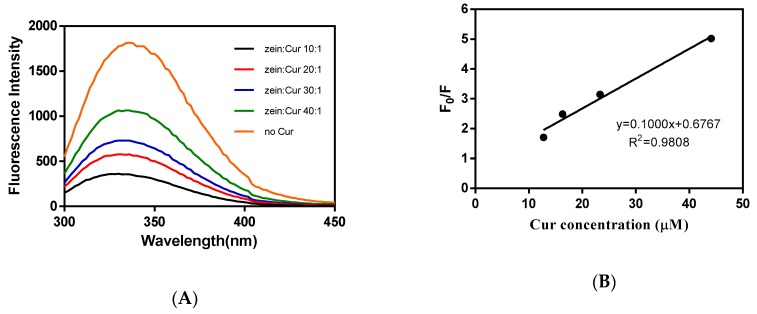
Changes in fluorescence emission spectra of blank zein-caseinate-pectin particles and Cur-loaded zein-caseinate-pectin particles with different zein to Cur mass ratios (**A**); linear relationship of F_0_/F versus Cur concentration in the Stern–Volmer equation (**B**).

**Figure 6 molecules-24-02535-f006:**
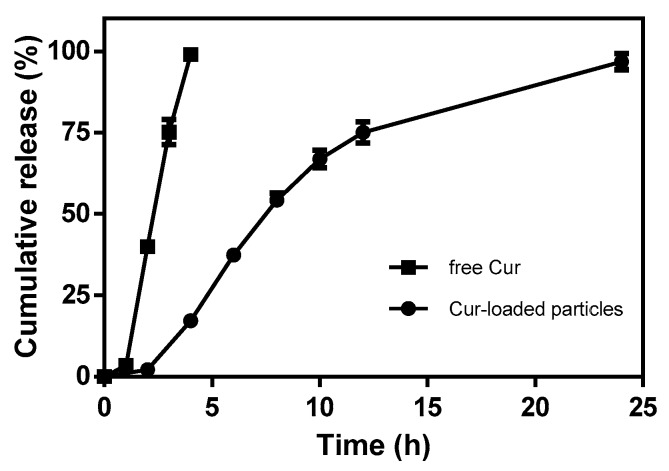
*In-vitro* release profiles of free Cur and encapsulated Cur from zein-caseinate-pectin particles.

**Figure 7 molecules-24-02535-f007:**
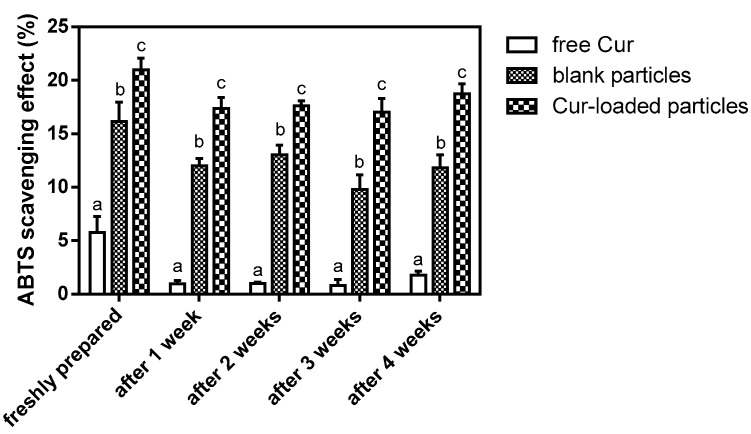
ABTS^•+^ radical scavenging capacity of the model beverage incorporated with free Cur, blank zein-caseinate-pectin particles or Cur-loaded zein-caseinate-pectin particles during storage at 4 °C. Values marked with different letters indicated a significant difference between different groups for the same storage time period (*p* < 0.05). Error bars represent standard deviations from three replicates.

**Figure 8 molecules-24-02535-f008:**
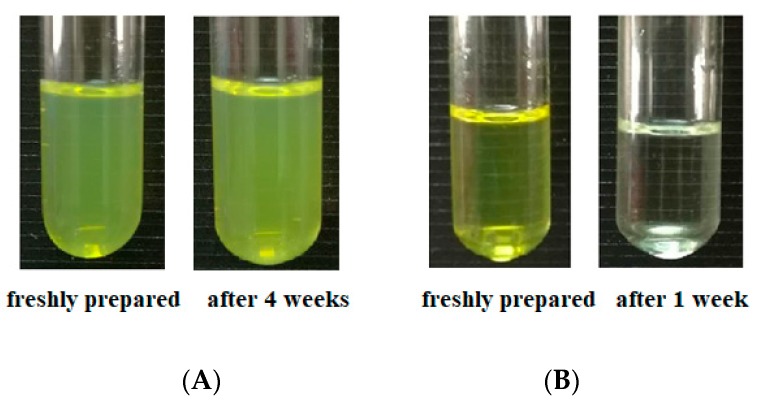
Visual appearance of the model beverage incorporated with Cur-loaded zein-caseinate-pectin particles (**A**) or free Cur (**B**) during storage at 4 °C.

**Table 1 molecules-24-02535-t001:** Characteristic data of zein-caseinate-pectin particles before and after lyophilization.

pH of the Dispersion	Z-Average (nm)		Polydispersity Index (PDI)		ζ-Potential (mV)	
	Freshly-Prepared	Re-Dispersion	Freshly-Prepared	Re-Dispersion	Freshly-Prepared	Re-Dispersion
3	291.6 ^a^ ± 4.0	385.0 ^a^ ± 4.7	0.2 ^a^ ± 0.0	0.2 ^a^ ± 0.0	−9.8 ^a^ ± 0.3	−23.4 ^a^ ± 0.6
3.5	344.2 ^b^ ± 3.9	407.2 ^a^ ± 6.0	0.2 ^a^ ± 0.0	0.2 ^a^ ± 0.0	−20.5 ^b^ ± 0.5	−26.3 ^b^ ± 0.8
4	358.3 ^b^ ± 3.5	513.9 ^b^ ± 13.5	0.2 ^a^ ± 0.0	0.1 ^a^ ± 0.0	−26.2 ^c^ ± 0.7	−32.9 ^c^ ± 0.8
4.5	359.7 ^b^ ± 7.8	510.3 ^b^ ± 10.5	0.2 ^a^ ± 0.0	0.2 ^a^ ± 0.0	−29.2 ^d^ ± 0.4	−31.4 ^c^ ± 0.7
5	386.9 ^c^ ± 8.6	542.1 ^c^ ± 3.1	0.2 ^a^ ± 0.0	0.2 ^a^ ± 0.0	−33.3 ^e^ ± 1.4	−37.0 ^d^ ± 0.5
6	405.7 ^d^ ± 8.1	573.1 ^d^ ± 6.3	0.2 ^a^ ± 0.0	0.2 ^a^ ± 0.0	−37.6 ^f^ ± 0.8	−41.6 ^e^ ± 0.9

^a^ Values are based on triplicate measurements. Mean ± SD values are shown. Different superscript lowercase letters in the same column indicate significant differences (*p* < 0.05).

**Table 2 molecules-24-02535-t002:** Characteristic data of the Cur-loaded zein-caseinate-pectin particles ^a^.

Zein:Cur (*w*/*w*)	Z-Average (nm)		Polydispersity Index (PDI)		ζ-Potential (mV)		Encapsulation Efficiency (%)
	Freshly-Prepared	Re-Dispersion	Freshly-Prepared	Re-Dispersion	Freshly-Prepared	Re-Dispersion	
10:1	365.7 ^a^ ± 3.3	489.7 ^a^ ± 8.6	0.2 ^a^ ± 0.0	0.2 ^a^ ± 0.0	−19.4 ^a^ ± 0.7	−27.4 ^a^ ± 0.3	81.3 ^a^ ± 0.4
20:1	358.4 ^a^ ± 6.9	509.5 ^a^ ± 4.4	0.2 ^a^ ± 0.0	0.2 ^a^ ± 0.0	−19.0 ^a^ ± 0.3	−28.0 ^a^ ± 0.3	86.0 ^b^ ± 0.2
30:1	366.2 ^a^ ± 4.8	497.4 ^a^ ± 10.1	0.2 ^a^ ± 0.0	0.2 ^a^ ± 0.0	−19.5 ^a^ ± 0.4	−27.8 ^a^ ± 0.3	89.8 ^c^ ± 0.3
40:1	369.2 ^a^ ± 1.4	496.0 ^a^ ± 8.3	0.2 ^a^ ± 0.0	0.2 ^a^ ± 0.0	−18.9 ^a^ ± 0.1	−28.1 ^a^ ± 0.4	94.0 ^d^ ± 0.8

^a^ Values are based on triplicate measurements. Mean ± SD values are shown. Different superscript lowercase letters in the same column indicate significant differences (*p* < 0.05).

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
