# Peer review of "Fabrication and Characterization of Zein Composite Particles Coated by Caseinate-Pectin Electrostatic Complexes with Improved Structural Stability in Acidic Aqueous Environments"

_molecules, 2019, doi:10.3390/molecules24142535_

Round 1
Reviewer 1 Report
The fabrication of Zein composite particles only with caseinate takes advantage of a low pH which increases their stability.Thus would be useful for acidic beverages.I beòeve less useful drug carrier in pharmaceutics. This should better sentenced in the discussion or in the conclusions.
The insertion in the particles of curcumin led to spherical shape with a narrow size distribution and a a negative surface charge.
As expected the encapsulated CUR interacted with carrier materials mainly through hydrophobic interactions as revrealed by Fluorescence spectroscopy. Stability during 4 weeks of storage and the radical scavenging capacity of curcumin is mantained.
The paper appears interesting as a carrier method for hydrophobic molecoles. It appears well conducted with the appropriate techniques and the results are well presented.
The only very small observation regard the strange abbreviations.
For Na cas I feel better SodiumCaseinate or SC and for Curcumin I suggest at least the term Cur instead of CUR .
Author Response
1. Reviewer #1: The fabrication of zein composite particles only with caseinate takes advantage of a low pH which increases their stability. Thus it would be useful for acidic beverages. I believe less useful for drug carrier in pharmaceutics. This should be better sentenced in the discussion or in the conclusions.
Response: No change has been made in the manuscript.
The potential application of zein-caseinate-pectin particles in acidic beverages has already been stated and highlighted in the Abstract and Conclusion Parts (Line 25-26 and 389-391).
2. Reviewer #1: The insertion in the particles of curcumin led to spherical shape with a narrow size distribution and a negative surface charge. As expected the encapsulated CUR interacted with carrier materials mainly through hydrophobic interactions as revealed by Fluorescence spectroscopy. Stability during 4 weeks of storage and the radical scavenging capacity of curcumin is maintained. The paper appears interesting as a carrier method for hydrophobic molecules. It appears well conducted with the appropriate techniques and the results are well presented. The only very small observation regards the strange abbreviations. For Nacas, I feel better Sodium Caseinate or SC and for Curcumin I suggest at least the term Cur instead of CUR.
Response: According to the reviewer’s suggestion, the abbreviation forms for Sodium Caseinate and Curcumin have been changed to caseinate and Cur, respectively. The changes have also been highlighted in red throughout the manuscript.
Reviewer 2 Report
Dear Authors,
the manuuscript titled "Fabrication and characterization of zein composite particles coated by caseinate-pectin electrostatic complexes..." reports on the encapsulation of curcumin by casein and pectin complexes in the presence of zein. The topic has been discussed in many other reports. Thus, the main problem of this work is that it is difficult to understand what is the original investigation and what is a mere confirmation of previous data.
For instance, the isoelectric point of zein (pH 6.2) or caseinate (pH 4.6) is well-known. So, the optimal pH chosen in the paragraph 2.1.1 is quite trivial and could be omited as this is not a new information.
Then, the fact that a more negative zeta potential is achieved by increasing the pectin concentration is also well-known and a consequence from the negative charge of pectin at lower pH. Thus, also paragraph 2.1.2 does not report any new insight.
Paragraph 2.1.3 reports a nice picture. However, this seems just summarizing the same results expressed in the previous two sections, without leading to any novelty.
PAragraph 2.1.4 is also well-known. The fact that pectin-zein particles have high stability by
maintaining their morphology before and after freeze-drying is, for instance, already reported in by Chang (10.1016/j.foodhyd.2017.03.033) or
Paragraph 2.15 fails to discuss the reasons for the higher re-dispersability observed before and after freeze drying. Previous works have already discussed this aspect. In particular, the higher dispersability was mainly associated with the use of ethanol in the preparation of caseinate solutions (Zhong, 10.1021/jf400752a). Instead, this paragraph seems just confirming what is already known, withoht offering any discussion on previous research.
As a further example, previous reports suggested that enhanced production of casein/pectin complexes can be achieved by heating the mixture at 85 °C for 30 min. This facilitates the formation of
stable, compact, and spherical nanocomplexes and seems superior to what has been done in the present manuscript (Luo, 10.1016/j.ijpharm.2015.03.043). In the previous work of Luo, it was demonstrated that heating not only greatly
increased the yield of nanocomplexes but also significantly improved the
encapsulation capability. However, this was ignored by the Authors.
Other works have already reported that pectin coating is an effective strategy to potentially realize practical applications of protein nanoparticles. For instance, PAn and Luo reported a PH-driven encapsulation technology for the self-assembled casein nanoparticles for enhanced dispersibility and bioactivity of of curcumin (10.1039/c4sm00239c). Chong provided insights into the formation and characterization of Zein/caseinate/pectin complex under different preparation conditions (10.1016/j.ijbiomac.2017.05.178). Overall, it is not clear what novelty is contained in the present work that is not already known.
For these reasons, I am sorry but I cannot recommend this work for publication.
Author Response
1. Reviewer #2: The manuscript titled "Fabrication and characterization of zein composite particles coated by caseinate-pectin electrostatic complexes..." reports on the encapsulation of curcumin by casein and pectin complexes in the presence of zein. The topic has been discussed in many other reports. Thus, the main problem of this work is that it is difficult to understand what is the original investigation and what is a mere confirmation of previous data. For instance, the isoelectric point of zein (pH 6.2) or caseinate (pH 4.6) is well-known. So, the optimal pH chosen in the paragraph 2.1.1 is quite trivial and could be omitted as this is not a new information.
Response: No change has been made in the manuscript.
The purpose of Section 2.1.1 is to choose an optimal pH value to facilitate the formation of caseinate-pectin electrostatic complexes, not to determine the isoelectric point of zein or caseinate. The pH value is an important factor that influences the structural stability of caseinate-pectin complexes and then the formation of zein-caseinate-pectin particles in this manuscript. Therefore, it’s necessary to optimize the pH value first and determine the formulation of zein-caseinate-pectin particles at the optimal pH value.
2. Reviewer #2: Then, the fact that a more negative zeta potential is achieved by increasing the pectin concentration is also well-known and a consequence from the negative charge of pectin at lower pH. Thus, also paragraph 2.1.2 does not report any new insight.
Response: No change has been made in the manuscript.
It is showed in Section 2.1.2 that the mass ratio of pectin to caseinate is an important factor affecting the stability of caseinate-pectin complexes and zein-caseinate-pectin particles at a fixed pH value of 3.5. Further increasing pectin to caseinate mass ratio from 1 to 2, the ζ-potential value was almost not changed and similar to that of pectin alone at pH 3.5, indicating the saturation of pectin molecules. Therefore, the data in Section 2.1.2 is necessary for optimizing the formulation of caseinate-pectin complexes and zein-caseinate-pectin particles. To clarify more clearly, Section 2.1.2 has been rewritten and highlighted in red (Line 78-89, 93-108).
3. Reviewer #2: Paragraph 2.1.3 reports a nice picture. However, this seems just summarizing the same results expressed in the previous two sections, without leading to any novelty.
Response: No change has been made in the manuscript.
Actually, the data in Section 2.1.3 is different from those in Section 2.1.1 and 2.1.2. In Section 2.1.3, zein-caseinate-pectin particles with the optimal formulation (2:1:1, w/w/w) were adjusted to a series of acidic pH values (pH 3-6) and determined for the acidic stability through visual inspection first, especially at pH values around the isoelectric point of caseinate (pH 4-5). Then, the particle size, particle distribution and ζ-potential values of zein-caseinate-pectin particles at these different acidic pH values were further measured. For Section 2.1.1 and 2.1.2, the formulation of zein-caseinate-pectin particles was optimized at a fixed pH value of 3.5 and only the ζ-potential value was measured.
4. Reviewer #2: Paragraph 2.1.4 is also well-known. The fact that pectin-zein particles have high stability by maintaining their morphology before and after freeze-drying is, for instance, already reported in by Chang (10.1016/j.foodhyd.2017.03.033).
Response: No change has been made in the manuscript.
The fabrication method for zein-caseinate-pectin complex particles in this article (10.1016/j.foodhyd.2017.03.033) was a pH- and heating-induced electrostatic adsorption method, which is different from that we used in this manuscript. Actually, this article has already been cited in this manuscript as reference 14 and relevant statement has been highlighted in the Introduction Part (Line 46-54). The physico-chemical properties of zein-caseinate-pectin particles fabricated by different methods may be different, so the re-dispersibility of zein-caseinate-pectin particles fabricated in this manuscript was also investigated.
5. Reviewer #2: Paragraph 2.15 fails to discuss the reasons for the higher re-dispersability observed before and after freeze drying. Previous works have already discussed this aspect. In particular, the higher dispersability was mainly associated with the use of ethanol in the preparation of caseinate solutions (Zhong, 10.1021/jf400752a). Instead, this paragraph seems just confirming what is already known, without offering any discussion on previous research.
Response: No change has been made in the manuscript.
In this manuscript, N2 is flowed to remove the ethanol from the system using a nitrogen evaporator, which has been stated and highlighted in Section 3.2 (Line 297-298). Besides, the storage stability of zein-caseinate-pectin particles was examined for measuring their changes in particle size, size distribution and ζ-potential value in Section 2.1.5, and it is not about the re-dispersibility of zein composite particles. To the best of our knowledge, few studies have reported the structural stability of zein-caseinate-pectin particles under different acidic pH conditions during storage.
6. Reviewer #2: As a further example, previous reports suggested that enhanced production of casein/pectin complexes can be achieved by heating the mixture at 85 °C for 30 min. This facilitates the formation of stable, compact, and spherical nanocomplexes and seems superior to what has been done in the present manuscript (Luo, 10.1016/j.ijpharm.2015.03.043). In the previous work of Luo, it was demonstrated that heating not only greatly increased the yield of nanocomplexes but also significantly improved the encapsulation capability. However, this was ignored by the Authors. Other works have already reported that pectin coating is an effective strategy to potentially realize practical applications of protein nanoparticles. For instance, Pan and Luo reported a PH-driven encapsulation technology for the self-assembled casein nanoparticles for enhanced dispersibility and bioactivity of curcumin (10.1039/c4sm00239c). Chong provided insights into the formation and characterization of Zein/caseinate/pectin complex under different preparation conditions (10.1016/j.ijbiomac.2017.05.178). Overall, it is not clear what novelty is contained in the present work that is not already known.
Response: No change has been made in the manuscript. Thank you for providing these three articles and one of the articles (10.1016/j.ijbiomac.2017.05.178) has already been cited in this manuscript as reference 15. For the article of 10.1039/c4sm00239c, an alkaline dissociation at pH 11 followed by acidification method was used to fabricate curcumin-loaded casein nanoparticles, while different particle compositions of zein, caseinate and pectin were used in this manuscript. Besides, the pH- and heating-induced electrostatic adsorption process was used for fabricating casein/pectin nanocomplexes (10.1016/j.ijpharm.2015.03.043) and zein/caseinate/pectin complex nanoparticles (10.1016/j.ijbiomac.2017.05.178). In this manuscript, it is shown that zein-caseinate-pectin particles with desirable acidic stability and encapsulation capacity could also be fabricated without heating treatment. Besides, different characterization methods for zein composite particles were used in this manuscript, such as the storage stability and antioxidant evaluation in model beverage systems during storage.
Reviewer 3 Report
The paper describes the preparation and characterization of zein coated nanoparticles. Formulation and stability studies have been carried on the nanoparticles, before and after curcumin encapsulation. It is my opinion the paper requires major revision before acceptance.
1) In the introduction chapter the authors state (line 55-56):"Thus, the primary purpose of this study was to fabricate zein composite particles stabilized by caseinate-pectin complexes using a simple and low energy cost method". I did not find any description of the method itself, nor comparison with previously described methods or either discussion concerning the benefits of the method. Which are, for example, the differences with the methods described in : Int J Biol Macromol 2017, 104 (A), 117-124 and/or LWT 2018, 89, 596-603.?
2) Why the release of curcumin has not been done? It would be the very first step approaching to nanoencapsulation of active compounds.
3) The HPLC method described for Curcumin analytical assessment is unclear, it should be better described.
4) Figure 4, concerning TEM analysis of curcumin loaded nanoparticles is too low resolution. It must be improved. Moreover TEM images of blank nanoparticles must be provided.
5) The paragraphs concerning the influence of mass ratio on nanoparticles are confusing, hard to follow and should be carefully rewritten.
6) The evaluation of the antioxidant activity by ABTS assay is inconsistent, in my opinion. How could encapsulated curcumin participate in a chemical reaction with ABTS? The concept of encapsulation is in sharp contrast with it. It is clear that the antioxidant properties of nanaparticles should be related to the polysaccharide or aminoacidic components, rather than curcumin. Please explain
Author Response
1. Reviewer #3: In the introduction chapter the authors state (line 55-56):"Thus, the primary purpose of this study was to fabricate zein composite particles stabilized by caseinate-pectin complexes using a simple and low energy cost method". I did not find any description of the method itself, nor comparison with previously described methods or either discussion concerning the benefits of the method. Which are, for example, the differences with the methods described in: Int J Biol Macromol 2017, 104 (A), 117-124 and/or LWT 2018, 89, 596-603.?
Response: A pH- and heating-induced complexation process was used to fabricate zein-caseinate-pectin particles in these two articles (Int J Biol Macromol 2017, 104(A), 117-124; LWT 2018, 89, 596-603), which relied on the thermal denaturation and aggregation of protein molecules as well as electrostatic interactions for complex particle formation (Line 46-54). In this manuscript, an electrostatic deposition and liquid-liquid dispersion method without heating treatment was used. To clarify more clearly, “the primary purpose of this study was to fabricate zein composite particles stabilized by caseinate-pectin complexes using a simple and low energy cost method” has been replaced with “Thus, the primary purpose of this study was to fabricate zein composite particles stabilized by caseinate-pectin complexes using an electrostatic deposition and liquid-liquid dispersion method without heating treatment, and the acidic stability of zein-caseinate-pectin composite particles was also investigated” (Line 55-58).
2. Reviewer #3: Why the release of curcumin has not been done? It would be the very first step approaching to nanoencapsulation of active compounds.
Response: No change has been made in the manuscript.
The release profile in simulated gastrointestinal fluids was commonly used to characterize the in vitro bioaccessibility of the encapsulated active compound. In this manuscript, the in vitro antioxidant activity of the encapsulated curcumin in model beverage system was evaluated instead, which might give more direct information on the potential application of zein composite particles for lipophilic nutrients in acidic beverages or drinks.
3. Reviewer #3: The HPLC method described for Curcumin analytical assessment is unclear, it should be better described.
Response: The detailed HPLC method for curcumin analytical assessment has been added (Line 336-344).
4. Reviewer #3: Figure 4, concerning TEM analysis of curcumin loaded nanoparticles is too low resolution. It must be improved. Moreover TEM images of blank nanoparticles must be provided.
Response: According to the reviewer’s comment, Figure 4 has been replaced with a new TEM image with a higher resolution (Page 7, Line 214). Besides, some statement about the TEM image has also been changed and highlighted on Line 205-208. Based on the data measured by dynamic light scattering technique (Table 1 and Table 2), there was no obvious changes on particle size, particle size distribution andζ-potential values of zein-caseinate-pectin particles before and after curcumin loading. Therefore, the TEM images of blank zein-caseinate-pectin particles were not taken.
5. Reviewer #3: The paragraphs concerning the influence of mass ratio on nanoparticles are confusing, hard to follow and should be carefully rewritten.
Response: The paragraphs concerning the influence of mass ratio of pectin to caseinate has been rewritten and highlighted on Line 78-89 and 93-108.
6. Reviewer #3: The evaluation of the antioxidant activity by ABTS assay is inconsistent, in my opinion. How could encapsulated curcumin participate in a chemical reaction with ABTS? The concept of encapsulation is in sharp contrast with it. It is clear that the antioxidant properties of nanoparticles should be related to the polysaccharide or amino acidic components, rather than curcumin. Please explain.
Response: No change has been made in the manuscript. Based on some previous reported literatures on the antioxidant evaluation of curcumin before and after encapsulation into nanoparticles (Colloids and Surfaces B: Biointerfaces, 2016, 148, 1-11; Journal of Controlled Release, 2011, 151, 176-182), the similar result that an improved free radical scavenging activity of curcumin in aqueous media after encapsulation into nanoparticles or nanospheres were observed. Besides, it is particularly noteworthy that the carrier material used in this article (Journal of Controlled Release, 2011, 151, 176-182) showed no radical scavenging activity, while the free radical scavenging activity of encapsulated curcumin can also be significantly improved comparing to the free curcumin. This result indicated that the antioxidant property of curcumin-loaded nanoparticles could be attributed to not only the carrier material of particles, but also the encapsulated curcumin, which was consistent with our observations in this manuscript. The fabricated zein composite particles coated by caseinate-pectin complexes not only provided nonpolar microenvironment for the encapsulated curcumin but also facilitated its dispersion in acidic aqueous environment, rendering a better contact probability with free radicals and a higher antioxidant activity than free curcumin.
Round 2
Reviewer 2 Report
Dear Author,
the manuscript has been revised according to the Reviewers comments. The resulting paper has been improved. It is scientifically sound and well organized. I have only few concerns about the quality of Tables and Figures. In details:
Concerning Tables, please, report the results according to the NIST guideline. Please, follows:
GLP 9 Rounding Expanded Uncertainties and Calibration Values (https://www.nist.gov/sites/default/files/documents/2019/05/14/glp-9-rounding-20190506.pdf).
In practice, in all Tables, round according to the second significant digits of the expression of uncertainty (For instance, in Table 2, the results 365.73 ± 3.27 should be expressed as 365.7 ± 3.3, and so on).
Concerning the Figures, the labels inside Figure 3 are too small. Please, keep size of labels readable, ideally, at the same size of the text in manuscript.
In Figure 5, the legend in the plot (A) is of poor quality. Difficult to read. In the plot (B), write the equation with consistent digits for the slope and intercept. In the caption of Figure 5, please, write in plain English the sentence "Linear plot of F0/F vs Cur".
Caption of Table 1. Change "lyophilizationa." in "lyophilization".
Caption of Figure 3. Explain in plain English the term "PDI".
Author Response
1. Reviewer #2: Concerning Tables, please, report the results according to the NIST guideline. Please follow: GLP 9 Rounding Expanded Uncertainties and Calibration Values (https://www.nist.gov/sites/default/files/documents/2019/05/14/glp-9-rounding-20190506.pdf). In practice, in all Tables, round according to the second significant digits of the expression of uncertainty (For instance, in Table 2, the results 365.73 ± 3.27 should be expressed as 365.7 ± 3.3, and so on).
Response: The data in all tables have been changed and highlighted in red (Line 135 and 190). Besides, the data throughout the manuscript have also been checked and changes have been highlighted in red (Line 83, 88, 96, 125-127, 148-149, 160-161, 185-186, 189, 253-254).
2. Reviewer #2: Concerning the Figures, the labels inside Figure 3 are too small. Please, keep size of labels readable, ideally, at the same size of the text in manuscript.
Response: The labels inside Figure 3 have been enlarged (Line 169).
3. Reviewer #2: In Figure 5, the legend in the plot (A) is of poor quality. Difficult to read. In the plot (B), write the equation with consistent digits for the slope and intercept. In the caption of Figure 5, please, write in plain English the sentence "Linear plot of F0/F vs Cur".
Response: The legends for different fluorescence emission spectra have been changed to solid lines with different colors (Figure 5, Line 235-236). Besides, the same four digits for the slope and intercept of the equation were shown in Figure 5B. “Linear plot of F0/F vs Cur” has been replaced with “Linear relationship of F0/F versus Cur concentration in the Stern-Volmer equation” (Line 238-239).
4. Reviewer #2: Caption of Table 1. Change "lyophilizationa." in "lyophilization".
Response: "lyophilizationa" has been replaced with "lyophilization" (Line 135).
5. Reviewer #2: Caption of Figure 3. Explain in plain English the term "PDI".
Response: “PDI” has been replaced with “polydispersity index (PDI)” (Line 20, 164, 175).
Reviewer 3 Report
The authors did not introduce substantial changes to the article. Nanoencapsulation is a widely diffused standard procedure but is meaningless when release studies are missing. Is curcumin released form nanoparticles or not? This is not a negligible aspect. Moreover, concerning antioxidant properties of encapsulated curcumin determined by ABTS I disagree with the procedure used, despite similar procedure has been used previously.In both paper cited by authors (Colloids and Surfaces B: Biointerfaces, 2016, 148, 1-11; Journal of Controlled Release, 2011, 151, 176-182) a release profile of curcumin is calculated and it could be expected that, during 30 minutes of reaction the released curcumin is responsible for the antioxidant activity. Finally, the low antioxidant effect of free curcumin at time zero is in sharp contrast with previosuly reported results.
Author Response
Reviewer #3: The authors did not introduce substantial changes to the article. Nanoencapsulation is a widely diffused standard procedure but is meaningless when release studies are missing. Is curcumin released form nanoparticles or not? This is not a negligible aspect. Moreover, concerning antioxidant properties of encapsulated curcumin determined by ABTS I disagree with the procedure used, despite similar procedure has been used previously. In both paper cited by authors (Colloids and Surfaces B: Biointerfaces, 2016, 148, 1-11; Journal of Controlled Release, 2011, 151, 176-182) a release profile of curcumin is calculated and it could be expected that, during 30 minutes of reaction the released curcumin is responsible for the antioxidant activity. Finally, the low antioxidant effect of free curcumin at time zero is in sharp contrast with previously reported results.
Response: Thank you for your comment and we also agree that the release profile of the encapsulated component is useful for food delivery system study. Based on a previous reported literature on a similar delivery system of curcumin-loaded cross-linked pectin-caseinate-zein complex nanoparticles, a sustained release profile was observed in simulated gastrointestinal fluids (Chang, C., Wang, T.R., Hu, Q. B., Luo, Y. C. (2017). Food Hydrocolloids, 72, 254-262). Therefore, the in vitro antioxidant activity of curcumin-loaded zein-caseinate-pectin particles was evaluated in our study instead of the release study. From the results in this manuscript, the curcumin-loaded nanoparticles showed a higher antioxidant activity than the free curcumin and blank nanoparticles during the 4 weeks storage, which also suggested that the encapsulated curcumin could be released from the nanoparticles. Furthermore, there are no nanoparticle samples left and the release study could not be added in this manuscript. In order to make up for the lacking of the release study, some discussion on the expected release profile and its relationship with the antioxidant activity was added in this revision and highlighted in red (Line 260-266). Also, the above mentioned article was cited as the reference 34 in this manuscript (Line 483-485).
At time zero, our results showed that the free curcumin with the concentration of 10 μg/mL had the ABTS scavenging activity of 5.78%. Based on two previously published literatures (Chang, C., Wang, T.R., Hu, Q. B., Luo, Y. C. (2017). Food Hydrocolloids, 72, 254-262; Chang, C., Wang, T.R., Hu, Q. B., Zhou, M. Y., Xue, J.Y., Luo, Y. C. (2017). Food Hydrocolloids, 70, 143-151), the reported ABTS scavenging activity of free curcumin with the same tested concentration was in the range of 7.5%-11%, which was comparable to that in our study. The slight difference in values might be due to the different dispersing system of free curcumin used. The ethanol aqueous system (80:20, v/v) and pure ethanol were used in this study and two previous literatures, respectively.